# TRANSFORMER-BASED OPEN-WORLD INSTANCE SEGMENTATION WITH CROSS-TASK CONSISTENCY REGULARIZATION

## ABSTRACT

Open-World Instance Segmentation (OWIS) is an emerging research topic that aims to segment class-agnostic object instances from images. The mainstream approaches use a two-stage segmentation framework, which first locates the candidate object bounding boxes and then performs instance segmentation. In this work, we instead promote a single-stage transformer-based framework for OWIS. We argue that the end-to-end training process in the single-stage framework can be more convenient for directly regularizing the localization of class-agnostic object pixels. Based on the transformer-based instance segmentation framework, we propose a regularization model to predict foreground pixels and use its relation to instance segmentation to construct a cross-task consistency loss. We show that such a consistency loss could alleviate the problem of incomplete instance annotation – a common problem in the existing OWIS datasets. We also show that the proposed loss lends itself to an effective solution to semi-supervised OWIS that could be considered an extreme case that all object annotations are absent for some images. Our extensive experiments demonstrate that the proposed method achieves impressive results in both fully-supervised and semi-supervised settings. Compared to SOTA methods, the proposed method significantly improves the $AP_{100}$ score by 4.75% in UVO dataset →UVO dataset setting and 4.05% in COCO dataset →UVO dataset setting. In the case of semi-supervised learning, our model learned with only 30% labeled data, even outperforms its fully-supervised counterpart with 50% labeled data. The code will be released soon.

## 1 INTRODUCTION

Traditional instance segmentation Lin et al. (2014); Cordts et al. (2016) methods often assume that objects in images can be categorized into a finite set of predefined classes (i.e., *closed-world*). Such an assumption, however, can be easily violated in many real-world applications, where models will encounter many new object classes that never appeared in the training data. Therefore, researchers recently attempted to tackle the problem of **Open-World Instance Segmentation (OWIS)** Wang et al. (2021), which targets class-agnostic segmentation of all objects in the image.

Prior to this paper, most existing methods for OWIS are of two-stage Wang et al. (2022); Saito et al. (2021), which detect bounding boxes of objects and then segment them. Despite their promising performances, such a paradigm cannot handle and recover if object bounding boxes are not detected. In contrast, a transformer-based approach called Mask2Former Cheng et al. (2022) has recently been introduced, yet only for closed-world instance segmentation. Based on the Mask2Former, we propose a Transformer-based Open-world Instance Segmentation method named TOIS.

Note that our work is not just a straightforward adaptation of Mask2Former from close-world to open-world. This is because unlike closed-world segmentation, where the object categories can be clearly defined before annotation, the open-world scenario makes it challenging for annotators to label all instances completely or ensure annotation consistency across different images because they cannot have a well-defined finite set of object categories. As shown in Figure 1(a), annotators miss some instances. **It still remains challenging that how to handle such incomplete annotations (i.e. some instances missed)**.

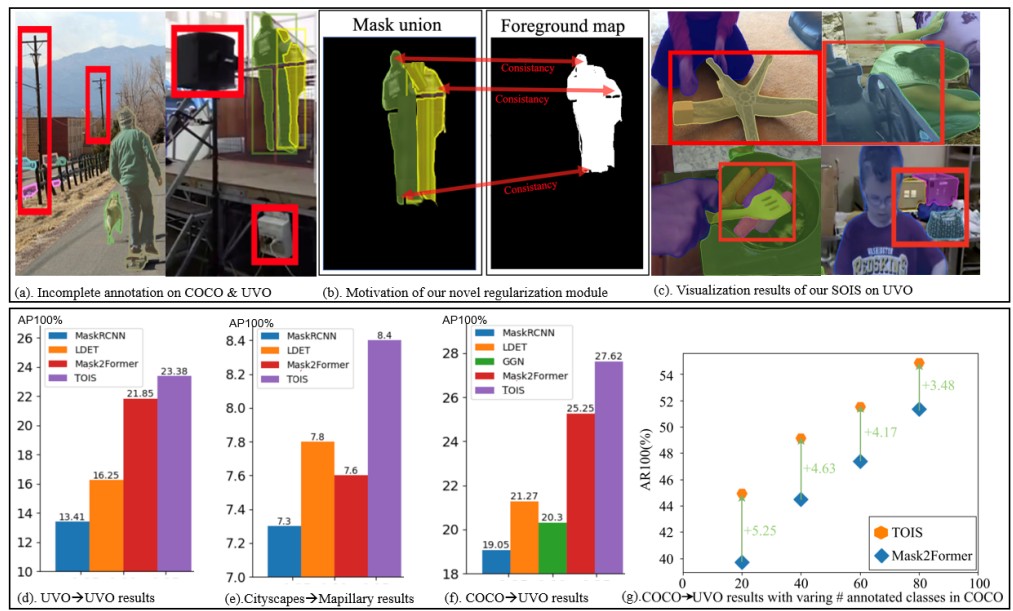

Figure 1: **(a).** Instances missing annotations in COCO and UVO datasets. The regions in red boxes are mistakenly annotated as background. **(b).** Motivation of our novel reg module (The consistency relationship between instance mask and foreground map). **(c).** Visualization results of our TOIS on UVO dataset. Here, the proposed TOIS is trained on COCO dataset and tested on UVO dataset. Our methods correctly segments many objects that are not labeled in COCO. **(d - f).** The $AP_{100}\%$ of our TOIS *vs.*SOTA methods on COCO→UVO, Cityscapes→Mapillary, COCO→UVO. **(g).** The $AR_{100}\%$ of our TOIS *vs.* baseline Mask2Former on COCO. From right to left, with the total number of classes decreases (i.e. more instance annotations missed), the gain of our TOIS over baseline becomes larger, thanks to the capability of our model to handle incomplete annotations.

Recent work LDET Saito et al. (2021) addresses this problem by generating synthetic data with a plain background, but based on a decoupled training strategy that can only be used in the two-stage method *while our method is of single-stage*. Another work called GGN Wang et al. (2022) handles such incomplete instance-level annotation issue by training a pairwise affinity predictor for generating pseudo labels. But training such an additional predictor is complicated and time-consuming.

In contrast, *our proposed TOIS method is end-to-end and simpler*. We address this incomplete annotation issue via a **novel regularization module**, which is simple yet effective. Specifically, it is convenient to concurrently predict not only (1) *instance masks* but also a (2) *foreground map*. Ideally, as shown in Figure 1(b), the foreground region should be consistent with the union of all instance masks. To penalize their inconsistency, we devise a *cross-task consistency loss*, which can down-weight the adverse effects caused by incomplete annotation. This is because when an instance is missed in annotation, as long as it is captured by both our predictions of instance masks and foreground map, the consistency loss would be low and hence encourage such prediction. Experiments in Figure 1(g) show that such consistency loss is effective even when annotations miss many instances. And as in Figure 1(c), novel objects which are unannotated in training set have been segmented successfully by our method.

So far, like most existing methods, we focus on the fully-supervised OWIS. In this paper, we further extend OWIS to the semi-supervised setting, where some training images do not have any annotations at all. This is of great interest because annotating segmentation map is very costly. Notably, **our proposed regularization module can also benefit semi-supervised OWIS** – consider an unlabeled image as an extreme case of incomplete annotation where all of the instance annotations are missed. Specifically, we perform semi-supervised OWIS by first warming up the network on the labeled set and then continuing training it with the cross-task consistency loss on the mixture of labeled and unlabeled images.

**Contributions**. In a nutshell, our main contributions could be summarized as:

1. Building upon a recently-proposed close-world segmentation method of Mask2Former, we propose a Transformer-based Open-world Instance Segmentation (TOIS) method.
2. We propose a novel cross-task consistency loss that mitigates the issue of incomplete mask annotations, which is a critical issue for open-world segmentation in particular.
3. We further extend the proposed method into a semi-supervised OWIS model, which effectively makes use of the unlabeled images to help the OWIS model training .
4. Our extensive experiments demonstrate that the proposed method reaches the leading OWIS performance in the fully-supervised learning. (Figure 1(d-f)), and that our semi-supervised extension can achieve remarkable performance with a much smaller amount of labeled data.

## 2 RELATED WORK

**Closed-world instance segmentation (CWIS)**    He et al. (2017); Chen et al. (2020); Dai et al. (2016); Bolya et al. (2019); Wang et al. (2020a) requires the approaches to assign a class label and instance ID to every pixel. Two-stage CWIS approaches, such as MaskRCNN, always include a bounding box estimation branch and a FCN-based mask segmentation branch, working in a 'detect-then-segment' way. To improve efficiency, one-stage methods such as CenterMask Dai et al. (2016), YOLACT Bolya et al. (2019) and BlendMask Chen et al. (2020) have been proposed, which remove the proposal generation and feature grouping process. To further free the CWIS from the local box detection, Wang et.al  Wang et al. (2020a) proposed SOLO and obtained on par results to the above methods. In recent years, the methods  Fang et al. (2021); Dong et al. (2021), following DETR Carion et al. (2020), consider the instance segmentation task as an ensemble prediction problem. In addition, Cheng et al. proposed an universal segmentation framework MaskFormer Cheng et al. (2021) and its upgrade version Mask2Former Cheng et al. (2022), which even outperforms the state-of-the-art architectures specifically designed for the CWIS task.

Notably, two-stage method CenterMask preserves pixel alignment and separates the object simultaneously by integrating the local and global branch. Although introducing the global information in this way helps improve the mask quality in CWIS, it cannot handle the open-world task very well, because CenterMask multiplies the local shape and the cropped saliency map to form the final mask for each instance. There is no separate loss for the local shape and global saliency. When such a method faces the incomplete annotations in OWIS tasks, the generated mask predictions corresponding to the unlabeled instances would still be punished during training, making it difficult to discover novel objects at inference. The efficient way to jointly take advantages of global and local information in OWIS tasks deserves to be explored.

**Open-world instance segmentation**    OWIS task Wang et al. (2021) here focuses on the following aspects: (1) All instances (without stuff) have to be segmented; (2) Class-agnostic pixel-level results should be predicted with only instance ID and incremental learning ability is unnecessary. Several OWIS works have recently been developed. Yu et al. Du et al. (2021) proposed a two-stage segmentation algorithm, which decoupled the segmentation and detection modules during training and testing. This algorithm achieves competitive results on the UVO dataset thanks to the abundant training data and the introduction of effective modules such as cascade RPN Vu et al. (2019), SimOTA Ge et al. (2021), etc. Another work named LDET Saito et al. (2021) attempts to solve the instance-level incomplete annotation problem. Specifically, LDET first generates the background of the synthesized image by taking a small piece of background in the original image and enlarging it to the same size as the original image. The instance is then matted to the foreground of the synthesized image. The synthesized data is used only to train the mask prediction branch, and the rest of the branches are still trained with the original data. Meanwhile, Wang et al. proposed GGN Wang et al. (2022), an algorithm that combines top-down and bottom-up segmentation ideas to improve prediction accuracy by generating high-quality pseudo-labels. Specifically, a Pairwise Affinity (PA) predictor is trained first, and a grouping module is used to extract and rank segments from predicted PA to generate pseudo-labels, which would be fused with groundtruth to train the segmentation model.

## 3 METHODOLOGY

In this section, we first define the OWIS problem with both fully and semi-supervised learning. Then the architecture of our TOIS and the proposed cross-task consistency loss are introduced in Section

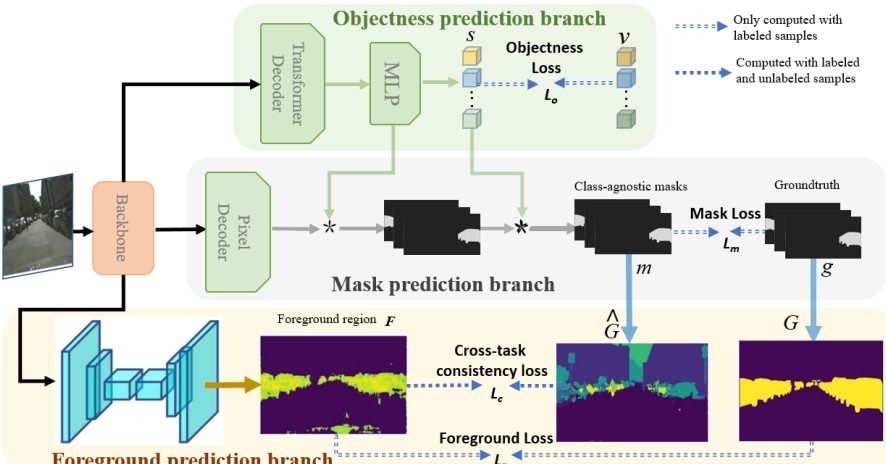

Figure 2: Overall framework of the proposed TOIS. The mask prediction branch generates the predicted masks, while the objectness prediction branch computes the objectness score for each mask. The foreground prediction branch segments a foreground region to guide the optimization of other two branches.

3.3 and 3.4, respectively. Finally, Section 3.4 and 3.5 show how to optimize the TOIS in fully and semi-supervised way, respectively.

## 3.1 PROBLEM DEFINITION OF OWIS

The open-world instance segmentation (OWIS) aims to segment all the object instances (things) of any class including those that did not appear in the training phase. Technically, OWIS is a task to produce a set of binary masks, where each mask corresponds to a class-agnostic instance. The pixel value of $1$ in the mask indicates a part of an object instance while $0$ indicates not.

## 3.2 MODEL ARCHITECTURE

Our proposed TOIS framework consists of three branches to alleviate the incomplete annotating, as shown in Figure 2. Basically, we follow the design of one-stage Mask2Former Cheng et al. (2022). The **objectness prediction branch** estimates the weighting score for each mask by applying a sequential Transformer decoder and MLP. The **mask prediction branch** predicts the binary mask for each instance. It first generates $N$ binary masks with $N$ ideally larger than the actual instance number $K_i$, which is the number of annotated object instances in a given image $i$. Each mask is multiplied by a weighting score with a value between $0$ and $1$, indicating if a mask should be selected as an instance mask. This process generates the mask in an end to end way, which avoids to miss the instance because of poor detection bounding boxes and meanwhile reduces the redundant segmentation cost for each proposal. We refer to Cheng et al. (2021; 2022) for more details.

The **foreground prediction branch** is a light-weight fully convolutional network to estimate the foreground regions that belong to any object instance.The more detailed design of the foreground prediction branch is in the Appendix. This guides the training of the mask branch through our cross-task consistency loss proposed in the following Sec. 3.3. Once training is done, we discard this branch and only use the objectness and mask prediction branch at inference time. Therefore, we would not introduce any additional parameter or computational redundancy, which benefits the running efficiency.

## 3.3 LEARNING WITH THE CROSS-TASK CONSISTENCY REGULARIZATION

A critical limitation of the OWIS is the never-perfect annotations due to the difficulties in annotating class-agnostic object instances. Towards alleviating this issue, we propose a regularization to provide extra supervision to guide the OWIS model training under incomplete annotations.

We construct a branch to predict the foreground regions that belong to any of the object instances. Formally we create the the foreground annotation $G(x, y)$ calculated by

$$G(x, y) = \begin{cases} 0, & \text{if } \sum_{i=1}^{K} g^i(x, y) == 0 \\ 1, & \text{otherwise,} \end{cases} \tag{1}$$

where $g^i(x, y)$ is one of the $K$ annotated object instances for the current image and the union of $g^i$ defines the foreground object regions. Here $(x, y)$ denotes a coordinate of a pixel in the an image. We use $G(x, y)$ as labels to train the foreground prediction branch.

Our consistency loss encourages the model outputs to have the relationship indicated in Eq 1, which states that the the foreground prediction should be the union of instance predictions. To do so, we use the following equation as an estimate of the foreground from the instance prediction:

$$\hat{G}(x, y) = \Phi \left( \sum_{j=1}^{K} m^j(x, y) \right), \tag{2}$$

where $m^j$ means the confidence of pixels in j-th predicted mask, and $\Phi$ represents the Sigmoid function. Then, let the foreground prediction from the foreground prediction branch be $F$, our cross-task consistency loss is to make $F$ and $\hat{G}(x, y)$ consistent, which finally leads to the following loss function.

$$L_c = \text{DICE}_{(\hat{G}, F)} + \text{BCE}_{(\hat{G}, F)}, \tag{3}$$

where DICE and BCE denote the dice-coefficient loss Milletari et al. (2016) and binary cross-entropy loss, respectively.

Consistency loss enjoys the following appealing properties. It is self-calibrated and independent with the incompleteness level of labels. As shown in Figure 3, for a instance mistakenly annotated as background, but the foreground prediction branch and mask prediction branch both correctly find it, the model would be punished through mask loss and foreground loss. However, the consistency loss thinks this prediction is correct. In this way, consistency loss down-weights the adverse effects caused by other unreliable segmentation losses. The mitigation and the compensation factor synergize to relieve the overwhelming punishments on unlabeled instances.

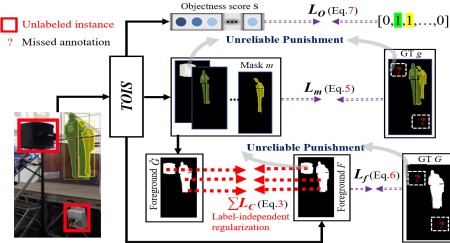

Figure 3: Working principle of consistency loss.

### 3.4 Fully-supervised learning

The overall fully-supervised optimization of the proposed TOIS is carried out by minimizing the following joint loss formulation $L_f$,

$$L_f = \alpha L_m + \beta L_p + \gamma L_c + \omega L_o, \tag{4}$$

$$\text{where} \quad L_m = \text{BCE}_{(m, g)} + \text{DICE}_{(m, g)}, \tag{5}$$

$$L_p = \text{BCE}_{(F, G)} + \text{DICE}_{(F, G)}, \tag{6}$$

$$L_o = \text{BCE}_{(s, v)}, \tag{7}$$

where $L_m$, $L_p$ and $L_o$ denote the loss terms for mask prediction, foreground prediction, and objectness scoring, respectively. $\alpha$, $\beta$, $\gamma$ and $\omega$ are the weights of the corresponding losses. $m$ and $g$ represent the predicted masks and corresponding groundtruth, respectively. $F$ and $G$ is the foreground prediction result and the generated foreground groundtruth, while the estimated objectness score is denoted with $s$. $v$ is a set of binary values that indicate whether each mask is an instance. Before computing the $L_m$, matching between the set of predicted masks and groundtruth has been done via the bipartite matching algorithm defined in Cheng et al. (2022).

### 3.5 EXTENSION TO SEMI-SUPERVISED LEARNING

Due to the ambiguity of the instance definition in OWIS, it is much harder for the annotators to follow the annotation instruction, and this could make the annotations for OWIS expensive. It is desirable if we can use unlabeled data to help train OWIS models. In this regard, our proposed cross-task consistency loss only requires the outputs of both predictors to have a consistent relationship indicated in Eq1, and does not always need ground truth annotations. Thus, we apply this loss to unlabeled data, which becomes semi-supervised learning. Specifically, the easier-to-learn foreground prediction branch is able to learn well through a few labeled images in the warm-up stage. Then the resulted foreground map can serve as a constraint to optimize the open-world mask predictions with the help of our cross-task consistency loss, when the labels do not exist. In this way, our Semi-TOIS achieves a good trade-off between the annotation cost and model accuracy.

**Semi-supervised learning process.** Given a labeled set $D_l = \{(x_i, y_i)\}_{i=1}^{N_l}$ and an unlabeled set $D_u = \{x_i\}_{i=1}^{N_u}$, our goal is to train an OWIS model by leveraging both a large amount of unlabeled data and a smaller set of labeled data. Specifically, we initially use $D_l$ to train the TOIS as a warm-up stage, giving a good initialization for the model. We then jointly train the OWIS model on the both labeled and unlabeled data. For the labeled data, we employ the loss function defined in Eq 4. For the unlabeled data, we apply only the cross-task consistency loss $L_c$.

## 4 EXPERIMENTS

For demonstrating the effectiveness of our proposed TOIS, we compared it with other fully-supervised methods through intra-dataset and cross-dataset evaluations. We also performed ablation studies in these two settings to show the effect of each component. Moreover, we apply the proposed cross-task consistency loss for semi-supervised learning and test our method on the UVO validation set.

### 4.1 IMPLEMENTATION DETAILS AND EVALUATION METRICS

**Implementation details**   Detectron2 Wu et al. (2019) is used to implement the proposed TOIS framework, multi-scale feature maps are extracted from the ResNet-50 He et al. (2016) or Swin Transformer Liu et al. (2021) model pre-trained on ImageNet Deng et al. (2009). Our transformer encoder-decoder design follows the same architecture as in Mask2Former Cheng et al. (2022). The number of object queries $M$ is set to 100. Both the ResNet and Swin backbones use an initial learning rate of 0.0001 and a weight decay of 0.05. A simple data augmentation method, Cutout DeVries & Taylor (2017), is applied to the training data. All the experiments have been done on 8 NVIDIA V100 GPU cards with 32G memory.

**Pseudo-labeling for COCO train set**   Pseudo-labeling is a common way to handle incomplete annotations. To explore the compatibility of our method and the pseudo-labeling operation, we employ a simple strategy to generate pseudo-labels for unannotated instances in the COCO train set Lin et al. (2014) in our experiments. Specifically, we follow a typical self-training framework, introducing the teacher model and student model framework to generate pseudo-labels. These two models have the same architecture, as shown in Figure 2, but are different in model weights. The weights of the student model are optimized by the common back-propagation, while the weight of the teacher model is updated by computing the exponential moving averages (EMA) of the student model. During training, the image $i$ is first fed into the teacher model to generate some mask predictions. The prediction whose confidence is higher than a certain value would be taken as a pseudo-proposal. The state $S_{ij}$ of the pseudo-proposal $p_{ij}$ is determined according to Equation (8).

$$
S_{ij} = \begin{cases} \text{True,} & \text{if } \text{argmax}(\varphi(p_{ij}, g_i)) \leqslant \varepsilon, \\ \text{False,} & \text{otherwise,} \end{cases} \tag{8}
$$

in which $g_i$ means any ground truth instance in the image $i$. $\varphi$ denotes the IOU calculating function, and $\varepsilon$ is a threshold to further filter the unreliable pseudo-proposals. Finally, pseudo-proposals with states $True$ would be considered as reliable pseudo-labels. Here, the confidence and IOU threshold $\varepsilon$ for selecting pseudo-labels are set to 0.8 and 0.2, respectively. Then, we jointly use the ground truth and the pseudo-labels to form the training data annotations. If a region is identified as belonging to an instance in the pseudo-label, it will be considered as a positive sample during training.

Table 1: Results of **UVO-train** → **UVO-val** intra-dataset evaluation.

| Metric | Backbone | $AP_{100}(\%)$ | $AP_s(\%)$ | $AP_m(\%)$ | $AP_l(\%)$ | $AR_{100}(\%)$ | $AR_{10}(\%)$ |
|---|---|---|---|---|---|---|---|
| MaskRCNN | R-50 | 13.41 | 4.91 | 12.33 | 17.45 | 22.77 | 20.01 |
| LDET | R-50 | 16.25 | 3.27 | 13.58 | 22.93 | 35.64 | 23.73 |
| Mask2Former | R-50 | 21.85 | 6.16 | 16.82 | 31.65 | 41.18 | 28.26 |
| **TOIS (Ours)** | R-50 | **23.38** | **6.59** | **17.35** | **34.23** | **41.94** | **29.24** |
| Mask2Former | Swin-B | 33.27 | 9.34 | 25.21 | 47.80 | 50.81 | 37.49 |
| **TOIS (Ours)** | Swin-B | **38.02** | **12.31** | **28.64** | **53.22** | **54.74** | **41.78** |

**Evaluation metrics**  The Mean Average Recall (AR) and Mean Average Precision (AP) Lin et al. (2014) are utilized to measure the performance of approaches in a class-agnostic way.

## 4.2 FULLY-SUPERVISED EXPERIMENTAL SETTING

**Intra-dataset evaluation**  UVO is the largest open-world instance segmentation dataset. Its training and test images are from the same domain, while they do not have any overlap. Here, we perform the learning process of TOIS on the UVO-train subset and conduct the test experiments on the UVO-val subset. Besides, we split the COCO dataset into 20 seen (VOC) classes and 60 unseen (none-VOC) classes. We train a model only on the annotation of 20 VOC classes and test it on the 60 none-VOC class, evaluating its ability of discovering novel objects.

Table 2: Results of **COCO2017-train(VOC)** → **COCO2017-val(none-VOC)** intra-set evaluation. $s, m$ and $l$ denote small, middle and large size of instances.

| Metrics | $AR_{100}(\%)$ | $AR_s(\%)$ | $AR_m(\%)$ | $AR_l(\%)$ |
|---|---|---|---|---|
| Mask2Former | 9.21 | 4.56 | 8.79 | 19.30 |
| TOIS | 11.03 | 4.87 | 9.24 | 26.81 |

**Cross-dataset evaluation**  Open-world setting assumes that the instance can be novel classes in the target domain. Therefore, it is essential for the OWIS method to handle the potential domain gap with excellent generalization ability. Cross-dataset evaluation, in which training and test data come from different domains, is necessary to be conducted. Here, we first train the proposed TOIS model and compare methods on the COCO-train subset, while testing them on the UVO-val dataset to evaluate their generalizability. Then we extend the experiments to an autonomous driving scenario, training the models on the Cityscapes Cordts et al. (2016) dataset and evaluating them on the Mapillary Neuhold et al. (2017). Cityscapes have 8 foreground classes, while Mapillary contains 35 foreground classes including vehicles, animals, trash can, mailbox, etc.

## 4.3 FULLY-SUPERVISED EXPERIMENTAL RESULTS

**Intra-dataset evaluation**  The results are illustrated in Table 1. The single-stage approaches based on the mask classification framework perform better than other two-stage methods. Among them, our proposed TOIS achieves a significant performance improvement over the Mask2Former baseline, which is 4.75% in $AP_{100}$ and 3.93% in $AR_{100}$ when using the Swin-B backbone. For VOC→none-VOC setting, the ex-

Table 3: Results of **COCO2017-train** → **UVO-val** cross-dataset evaluation.

| Metric | Backbone | $AR_{100}(\%)$ | $AP_{100}(\%)$ | $AP_s(\%)$ | $AP_m(\%)$ | $AP_l(\%)$ |
|---|---|---|---|---|---|---|
| B-MaskRCNNCheng et al. (2020) | R-50 | 36.21 | 20.40 | 5.98 | 13.16 | 30.69 |
| Mask TransfinerKe et al. (2022) | R-50 | 36.41 | 21.51 | 6.71 | 14.20 | 32.12 |
| BCnetKe et al. (2021) | R-50 | 36.28 | 20.87 | 6.42 | 13.92 | 31.27 |
| PointRendKirillov et al. (2020) | R-50 | 37.02 | 20.40 | 8.99 | 15.81 | 34.57 |
| MaskRCNNHe et al. (2017) | R-50 | 38.17 | 19.05 | 6.27 | 13.15 | 28.05 |
| LDETSaito et al. (2021) | R-50 | 42.63 | 21.27 | 5.66 | 17.52 | 18.38 |
| GGNWang et al. (2022) | R-50 | 43.30 | 20.30 | **8.70** | 18.20 | 27.30 |
| Mask2FormerCheng et al. (2022) | R50 | 48.71 | 25.24 | 6.46 | 16.09 | 40.37 |
| **TOIS (Ours)** | R-50 | **51.28** | **27.62** | 7.80 | **18.61** | **43.42** |
| Mask2Former | Swin-B | 51.38 | 28.16 | 7.29 | 18.91 | 45.48 |
| **TOIS(Ours)** | Swin-B | **54.86** | **32.21** | **9.03** | **21.92** | **50.69** |

perimental results are shown in Table 2, which verified that our proposed method can improve the performance for all instances, especially large ones.

**Cross-dataset evaluation**  For the COCO→UVO task, according to Table 3, it is clear that the proposed TOIS outperforms all previous methods, achieving a new state-of-the-art $AR_{100}$ at 54.86% which is 11.56% higher than previous state-of-the-art method GGN Wang et al. (2022). We also applied the proposed techniques to another classic one-stage method SOLO V2 Wang et al. (2020b). The experimental results in Table 4 show that it improves $AR_{100}$ and $AP_{100}$ by 3.11% and 2.79% compared to SOLO V2. For the Cityscapes→ Mapillary task, the overall $AP$ and $AR$ of TOIS still surpass the performance of other state-of-the-art methods ( in Table 5 ), which demonstrates the

Table 4: Results of TOIS with SOLOV2 structure ( **UVO-train**→ **UVO-val**).

| Metric | Backbone | $AR_{100}(\%)$ | $AP_{100}(\%)$ | $AP_s(\%)$ | $AP_m(\%)$ | $AP_l(\%)$ |
|---|---|---|---|---|---|---|
| SOLO V2 | R-50 | 39.41 | 22.25 | 5.56 | 14.18 | 34.12 |
| SOLO V2TOIS | R-50 | 42.52 | 25.04 | 6.77 | 16.90 | 38.33 |

Table 5: Cross-dataset evaluation on autonomous driving scenes. Results of **Cityscapes** → **Mapillary.**

| Method | MaskRCNN | LDET | Mask2Former | OSIS(Ours) |
|---|---|---|---|---|
| AP(%) | 7.3 | 7.8 | 7.6 | 8.4 |
| $AR_{10}(\%)$ | 6.1 | 5.5 | 7.0 | 7.5 |

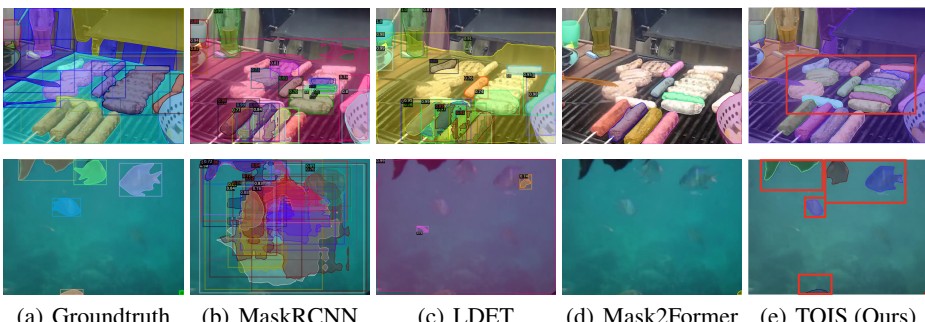

(a) Groundtruth    (b) MaskRCNN    (c) LDET    (d) Mask2Former    (e) TOIS (Ours)

Figure 4: Visualization results of COCO→UVO cross-dataset evaluation. The predicted boxes of two-stage methods MaskRCNN and LDET are also drawn. Proposed TOIS can discover both unlabeled objects (first row) and unseen class of instances (second row) as shown in red boxes.

effectiveness of our proposed techniques. We show some of the COCO→UVO visualization results in Figure 4 to qualitatively demonstrate the superiority of our method. Please refer to the supplementary material for more qualitative examples.

## 4.4 ABLATION STUDY

We perform cross-dataset and intra-dataset ablation studies to analyze the effectiveness of each component in the proposed TOIS, including the foreground prediction branch and the cross-task consistency loss. We also try combinations of the pseudo-label generation strategy and our cross-task consistency loss to investigate the individual and synergetic effects of them. Using the SwinB backbone, these models are trained on the COCO-train subset and the UVO-train subset, respectively. The metrics reported in Table 6 are tested on the UVO-val dataset.

**Effectiveness foreground prediction branch**
Table 6 shows that although a separate foreground prediction branch can guide the method to optimize towards the direction of discovering foreground pixels, it only slightly boosts the performance.

**Effectiveness of cross-task consistency loss**
Cross-task consistency loss has a positive effect on both sparse annotated (COCO) and dense annotated (UVO) training dataset. The values of $AP_{100}$ and $AR_{100}$ increase significantly ( 2.74% ↑ and 2.49% ↑ on COCO while 2.90% ↑ and 3.35%↑ on UVO) after applying the cross-task consistency loss as well as the foreground

Table 6: Ablation results of the proposed components by cross-dataset and intra-dataset evaluations. Foreground prediction (FP), Cross-task consistency (CTC) loss, Pseudo label (PL).

| Component | | | Train on COCO | | Train on UVO | |
|---|---|---|---|---|---|---|
| FP | CTC loss | PL | $AP_{100}(\%)$ | $AR_{100}(\%)$ | $AP_{100}(\%)$ | $AR_{100}(\%)$ |
| | | | 28.65 | 51.54 | 35.12 | 51.39 |
| ✓ | | | 29.02 | 51.60 | 35.55 | 51.73 |
| | | ✓ | 30.09 | 52.97 | 32.94 | 51.64 |
| ✓ | ✓ | | 31.39 | 53.83 | **38.02** | **54.74** |
| ✓ | | ✓ | 30.17 | 52.98 | 33.35 | 50.90 |
| ✓ | ✓ | ✓ | **32.21** | **54.86** | 37.71 | 52.27 |

prediction branches together. This result outperforms the TOIS counterpart with only pseudo-labeling, showing our effectiveness. In addition, jointly utilizing our cross-task consistency loss as well as the pseudo-labeling strategy leads to performance improvements on two settings, which demonstrates the synergistic effect of both approaches.

**Effectiveness of pseudo-labeling** Pseudo-labeling is not always necessary and powerful for any types of datasets. As shown in Table 6, the $AP_{100}$ and $AR_{100}$ of the COCO trained model increase by 1.35% and 0.78%, respectively, after applying the pseudo-label generation. However, pseudo-labeling causes a performance degradation (e.g. 2.18%↓ in $AP_{100}$) to a model trained in the UVO dataset. Compared with COCO, the UVO dataset is annotated more densely. We conjecture that the

Table 7: Results of our TOIS and classic semi-supervised method on UVO-val.

| Training Data | UVO-train with 30% annotation | | | |
|---|---|---|---|---|
| Method | Fully-TOIS$_{30}$ | Mean Teacher | Pseudo Labeling | Semi-TOIS$_{30}$ |
| AP$_{100}$(%) | 21.67 | 21.95 | 22.77 | 25.03 |
| AR$_{100}$(%) | 40.09 | 40.82 | 41.56 | 45.42 |

Table 8: Results of our TOIS and recent end to end method on UVO-val.

| Training Data | UVO-train with 50% annotation | | | |
|---|---|---|---|---|
| Method | LDET$_{50}$ | Mask2Former$_{50}$ | Fully-TOIS$_{50}$ | Semi-TOIS$_{50}$ |
| AP$_{100}$(%) | 10.61 | 19.49 | 22.86 | 25.22 |
| AR$_{100}$(%) | 25.08 | 38.08 | 41.44 | 47.56 |

background annotations of UVO are more reliable than those of COCO, where carefully selected pseudo-labels are more likely to represent unlabeled objects. The generated pseudo-labels of UVO contain higher noises than those of COCO. These additional noisy labels mislead the model training.

### 4.5 SEMI-SUPERVISED LEARNING EXPERIMENT

**Experimental setting** We have divided the UVO-train dataset into the labeled subset $D_l$ and the unlabeled subset $D_u$. Semi-supervised model Semi-TOIS is optimized as described in Section 3.5 on $D_l \cup D_U$, while the fully-supervised method Labelled-Only-TOIS is trained merely on the $D_L$. To ensure the comprehensiveness of the experiments, two different data division settings are included in our experiments:$\{D_L=30\%, D_u=70\%\}$ and $\{D_L=50\%, D_u=50\%\}$. The backbone applied here is Swin-B. We also implemented the classic Mean teacher model and a simple pseudo-label method based on the Mask2Former to perform comparison. Note that our semi-supervised setting is different from that in some previous work, like in MaskXRCNNHu et al. (2018) and ShapemaskKuo et al. (2019). They assume that the training set $C = A \cup B$, where examples from the categories in $A$ have masks, while those in $B$ have only bounding boxes.

**Results and analyss** As presented in Figure 5, the Semi-$TOIS_{50}$ model trained on the UVO with 50% annotated data outperforms the Semi-$TOIS_{30}$ model learning with 30% labeled training images. However, the performance increase between the Semi-$TOIS_{30}$ and Semi-$TOIS_{50}$ is slight. In addition, Semi-$TOIS_{30}$ improves Labelled-Only-$TOIS_{30}$ by 3.36% and 5.33% in $AP_{100}$ and $AR_{100}$, respectively. Compared to Labelled-Only-$TOIS_{50}$, Semi-$TOIS_{50}$ still achieves significant advantages (2.36% in $AP_{100}$ and 6.12% in $AR_{100}$). These results reflect that cross-task consistency loss has the ability to extract information from unlabeled data and facilitates model optimization in the semi-supervised setting. It is notable that the results of Semi-$TOIS_{30}$ are even better than those of Labelled-Only-$TOIS_{50}$. This illustrates that the information dug out by the cross-task consistency loss from the remaining 70% unlabeled data is more abundant than that included in 20% fully-labeled data. Therefore, our algorithm can achieve better performance with fewer

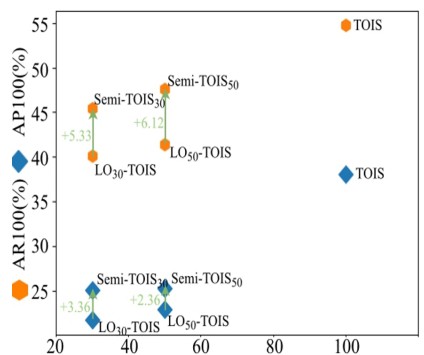

Figure 5: Comparison between TOIS, Labelled-Only-TOIS(LO-TOIS) and semi-TOIS.

annotations. This characteristic is promising in solving the OWIS problem. In addition, we also compared the semi-$TOIS$ with classic semi-supervised method and recent end to end segmentation method. The results in Table 7 and 8 show our advantages over the compared methods.

## 5 CONCLUSION

This paper proposes the first transformer-based framework (TOIS) for the open-world instance segmentation task. Apart from predicting the instance mask and objectness score, our framework introduces a foreground prediction branch to segment the regions belonging to any instance. Utilizing the outputs of this branch, we propose a novel cross-task consistency loss to enforce the foreground prediction to be consistent with the prediction of the instance masks. We experimentally demonstrate that this mechanism alleviates the problem of incomplete annotation, which is a critical issue for open-world segmentation. Our extensive experiments demonstrate that TOIS outperforms state-of-the-art methods by a large margin on typical datasets. We further demonstrate that our cross-task consistency loss can utilize unlabeled images to obtain some performance gains for a semi-supervised instance segmentation. This is an important step toward reducing laborious and expensive human annotation.

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

## A APPENDIX

Please refer to the additional supplement material.

