# OpenReview forum: "Single-Stage Open-world Instance Segmentation with Cross-task Consistency Regularization"
_ICLR.cc/2023/Conference — Submitted to ICLR 2023_

### Official Review · Reviewer_baa2 · 2022-10-22

**Confidence:** 5
**Correctness:** 3
**Technical Novelty And Significance:** 2
**Empirical Novelty And Significance:** 3
**Recommendation:** 6

**Clarity, Quality, Novelty And Reproducibility:**

Clarity: Mostly clear, but could be improved further by addressing the points noted in the weaknesses section above.

Quality: Meets the quality standards of an ICLR submission in terms of the technical contribution and presentation.

Novelty: Sufficiently novel in both the proposed method and the fairly open problem addressed.

Reproducibility: The method is well described. The supplementary provides the implementation details. The authors provide the code and have promised to release it.

**Strength And Weaknesses:**

Strengths:
Novelty: This paper tackles the problem of open-world instance segmentation, which is a largely open, albeit an important problem in computer vision research. If solved it has huge implications in being able to go from successfully segmenting a few hundred categories of objects today to potentially infinite ones in the future. Hence, it is important to further develop this line of research.

In terms of the proposed method itself, there are several novel aspects to it. The authors are the first to apply the SOTA transformer-based Mask2former architecture to the task of open-world instance segmentation. Given the known success of the Mask2former in the closed-world setting it is not surprising that also helps to significantly boost accuracies for the proposed method in the open-world setting. Hence, I consider this as a somewhat incremental contribution. Nevertheless, it is to the authors' credit that they are the first to numerically show this to be true.

The more interesting aspects of the authors' method, however, are the proposed foreground prediction branch and the self-supervised cross-task loss associated with it, which are designed to deal with the problem of "missing annotations" in the COCO dataset. I found this design to be clever, task-specific, conceptually correct and numerically shown to help in boosting the accuracy for open-world instance segmentation.

Significance of results:
In absolute terms the proposed method helps to significantly improve the accuracy of the open-world instance segmentation task. That said, the major boost comes from using Mask2Former and less from the other components. Nevertheless, this work is significant in advancing accuracy of this task. Additionally the proposed background prediction module is shown to be multi-purpose. It can be used both for open-world segmentation and for semi-supervised training.

Weaknesses
Claims:
1. The authors claim in the introduction "We propose a Singe-stage Open-world Instance Segmentation (SOIS) for the first time". This is technically not true. The previous GGN work is also a single-stage method based on SOLOv2. In relation to the existing work, the novel contribution of this work is to be the first to apply a transformer-based Mask2Former architecture to the task of open-world instance segmentation. The authors should correct their claims appropriately.

Experiments:
2. How sensitive is the proposed method's success to choosing the appropriate weight for the $L_c$ loss term to negate the effect of the network incorrectly penalizing missing detections? Does it need to be set differently for the different datasets?

3. In Table 1, what is the intuition behind the proposed foreground prediction loss helping in the intra-dataset setting? If it is mostly designed to catch un-annotated detections, how does it help to segment the labeled annotations?

4. Section 4.5.
It would be better to call the Fully-SOIS to "Labelled-Only-SOIS". The term "full" seems to imply that the full training set of OVU was used versus just the labeled 30%/50% set. Related to this point, in figure 5, for completeness the authors should also show the results of training with the full OVU training set, ie., the numbers from Table 6.

Minor points:
1. Introduction, Para 2: "which detects" --> "which detect"
2. In Fig.2, the masks don't correspond to the input image at all. Although they are meant to show the concept, nevertheless it would be better to at least show masks that are somewhat representative of the input scene and not those of faces!
3. In several places, "leaning" --> "learning"
4. Table 2, clarify what $s$, $m$ and $l$ mean.

**Summary Of The Paper:**

This paper proposes a method for single-stage open-world instance segmentation. It is the first to uses the SOTA Mask2Former architecture for this task. Additionally, to tackle the problem of "missing annotations" while training for open-world segmentations with the limited category labels in the training dataset, the authors propose a foreground prediction branch along with a cross-task consistency loss. The authors further show that since the latter loss does not require any labels, it can also be effectively employed  to train networks in a semi-supervised manner. The authors show quantitatively the superior accuracy of their approach in comparison to the existing state-of-the-art approaches on Open-World instance segmentation, on several baseline datasets. Additionally, they also show the contribution of each component of their design via ablation studies and the usefulness of their proposed approach in semi-supervised learning.

**Summary Of The Review:**

Overall the paper makes a significant contribution in advancing the field of open-world instance segmentation. Hence I am leaning towards accepting this paper, with the caveat that the authors improve the claims and clarity of their paper as noted in the "weaknesses" section above.

---

> ### Author Response · Authors · 2022-11-18
> **Response to Reviewer baa2**
>
> **We would like to thank the referee again for taking the time to review our manuscript.**
>
> **Q1: The authors claim in the introduction "We propose a Singe-stage Open-world Instance Segmentation (SOIS) for the first time". This is technically not true. The previous GGN work is also a single-stage method based on SOLOv2. In relation to the existing work, the novel contribution of this work is to be the first to apply a transformer-based Mask2Former architecture to the task of open-world instance segmentation. The authors should correct their claims appropriately.**
>
> **A1:** Thank you for the suggestion. For the experimental results of the single-stage algorithm are not provided in the GGN paper, we have emphasized that our algorithm is the first single-stage OWIS algorithm. We agree that GGN can be adapted for single-stage baseline as well. Therefore, we follows the reviewer's suggestion and adopted a rigorous statement in the revised paper: "we present the first transformer-based OWIS method."
>
> **Q2: How sensitive is the proposed method's success to choosing the appropriate weight for the loss term L_c to negate the effect of the network incorrectly penalizing missing detections? Does it need to be set differently for the different datasets?**
>
> **A2:** The weights of consistency loss are important hyper-parameters related to the incompleteness of the dataset annotation. The more the missing instance annotations of the dataset, the higher the weight of consistency loss. Due to the limited space, we have provided the weights of consistency loss for different datasets in Section A.1 in our revised supplementary.
>
> **Q3: In Table 1, what is the intuition behind the proposed foreground prediction loss helping in the intra-dataset setting? If it is mostly designed to catch un-annotated detections, how does it help to segment the labeled annotations?**
>
> **A3:** Ideally, all the instances in UVO have been labeled. However, as seen from the example images in Figure 7 in supplementary, *there are still some unannotated instances in UVO* . This is largely attributed to the challenge of annotating OWIS datasets. Unlike close-world segmentation tasks, the definition of 'object instance' in OWIS is much more diverse and ambiguous. Consequently, the annotators often find it hard to follow the annotation instructions and thus make much more mistakes. Therefore, although designed to catch unannotated instances, our methods also work on inner-dataset evaluations on UVO.
>
> **Q4: Section 4.5. It would be better to call the Fully-SOIS to "Labelled-Only-SOIS". The term "full" seems to imply that the full training set of UVO was used versus just the labeled 30\%/50\% set. Related to this point, in figure 5, for completeness the authors should also show the results of training with the full UVO training set, ie., the numbers from Table 6.**
>
> **A4:** Thanks for kindly point out this, we have revised Figure 5 accordingly.
>
> **Q5: Minor points: Introduction, Para 2: "which detects" --> "which detect"; In Fig.2, the masks do not correspond to the input image at all. Although they are meant to show the concept, nevertheless it would be better to at least show masks that are somewhat representative of the input scene and not those of faces! In several places, "leaning" --> "learning"; Table 2, clarify what s,m and l  and  mean.**
>
> **A5:** Appreciate for suggestion, we have corrected them and added the corresponding explanation.

---

> > ### Comment · Reviewer_baa2 · 2022-11-28
> > **Response to Authors and Final Score**
> >
> > I thank the authors to addressing all my concerns and updating the paper. However, after carefully reviewing all the feedback from all authors, I have to agree with the consensus among the reviewers that technical novelty of the paper is weak. It is mostly in adapting the MaskFormer architecture to the OWIS task and in proposing the consistency regularization. However, the majority of the absolute gain of the proposed method comes from using the Mask2Former architecture versus the regularization loss. Overall, while I still think that the empirical contribution of the paper is good in the absolute sense of advancing the field, the technical novelty in solving the OWIS task is incremental. Hence, I have lowered my final score to marginally above the acceptance level.

---

> > > ### Author Response · Authors · 2022-12-11
> > > **Response to Reviewer baa2**
> > >
> > > **We gratefully thanks for the precious time the reviewer spent making constructive remarks.**
> > >
> > > **Q1:  The majority of the absolute gain of the proposed method comes from using the Mask2Former architecture versus the regularization loss.**
> > >
> > > **A1:** We thank the reviewer again for providing us with useful comments.  Based on the literature data, we respectfully disagree with this point. The introduction of the Mask2Former framework does boost the performance. Meanwhile, as shown in the Table 1 and Table 2, **the performance gain introduced by the consistency loss is significant and comparable to the excellent SOTA work.**
> > >
> > > *Table 1：AP100% improvements of some SOTA work compared to their baselines on COCO→UVO setting.*
> > >
> > > | Method         | Baseline    | AP100%_{method}-AP100%_{Baseline} |
> > > |----------------|-------------|-------------------------|
> > > | LDET(ECCV2022) | MaskRCNN    | 2.22                    |
> > > | GGN(CVPR2022)  | MaskRCNN    | 1.25                    |
> > > | Ours           | Mask2Former | 4.05                    |
> > >
> > >
> > > *Table 2：AP100% improvements of some SOTA work compared to their baselines on UVO→UVO setting.*
> > > | Method         | Baseline    | AP100%_{method}-AP100%_{Baseline} |
> > > |----------------|-------------|-------------------------|
> > > | LDET(ECCV2022) | MaskRCNN    | 2.84                    |
> > > | Ours           | Mask2Former | 4.75                    |

---

### Official Review · Reviewer_THoq · 2022-10-24

**Confidence:** 4
**Clarity, Quality, Novelty And Reproducibility:** Good paper structure. Figure 5 is blu…
**Correctness:** 3
**Technical Novelty And Significance:** 2
**Empirical Novelty And Significance:** 2
**Recommendation:** 6

**Strength And Weaknesses:**

Pros:

1. The paper proposes a single-stage open-world Instance segmentation framework based on Mask2Former.
2. Adding an auxiliary task and a consistency loss for self-supervised learning, which improves a lot on both UVO-to-UVO and COCO-to-UVO settings.
3. Experiments demonstrate the semi-supervised framework can achieve remarkable performance with much smaller amount of labeled data.

Cons:
1. Although the paper proposed a new constraint called “cross-task consistency loss” for OWIS problem, it lacks of sufficient validation by either theoretical interpretation or experimental evidence how the cross-task consistency loss works for providing an error-correcting mechanism for handling noisy annotations. How does the “cross-task consistency loss” help to handle and separate neighboring occlusion objects?

2. The tech novelty of the paper is weak and incremental. Being first to adapt Mask2Former to OWIS is an incremental idea. Also, it should be called query-based or transformer-based, instead of one-stage. Can the author give proof on why the auxiliary task of predicting total foreground mask will help novel object detection?

3. Missing comparison/discussion with existing semi-supervised instance segmentation methods, such as [a, b].

[a] Learning to segment every thing. CVPR, 2018.

[b] Shapemask: Learning to segment novel objects by refining shape priors. ICCV, 2019.

4. Missing two-stage works on Closed-world instance segmentation, such as [c, d, e].

[c] PointRend: Image Segmentation as Rendering. CVPR, 2019.

[d] Deep occlusion-aware instance segmentation with overlapping bilayers. CVPR, 2021.

[e] Boundary-preserving Mask R-CNN. ECCV, 2020.

**Summary Of The Paper:**

This paper proposes an open world instance segmentation framework, which contains three stages: 1) it proposes an auxiliary task of predicting total foreground on the basis of existing query-based mask2former model. 2) it includes a cross-task consistency loss and improve results by correcting outputs between two tasks. 3) the paper also designs a way of adding pseudo labels on COCO, which gain some improvement on open-world settings, including semi-supervised setting.

**Summary Of The Review:**

Limited technical novelty. Deeper theoretical analysis is desired. My rating will be adjusted according to the author response.

---

> ### Author Response · Authors · 2022-11-18
> **Response to Reviewer THoq (Q5-Q7)**
>
> **Q5: Can the author give proof on why the auxiliary task of predicting total foreground mask will help novel object detection?**
>
> **A5:** Ablation results in Table 6 show that the introduction of the foreground prediction task alone brings no more than 1\% improvements, Further significant improvement comes from bringing in consistency loss.
>
> **Q6: Missing comparison/discussion with existing semi-supervised instance segmentation methods, such as [a, b] and missing two-stage works on Closed-world instance segmentation, such as [c, d, e].**
>
> **A6:** Thanks for providing such excellent work to improve the experiments of this paper. We have added [c][d][e] in our experiments, as shown in Table 3 in the revised paper.
>
> ### ##The results of new added compared methods on COCO→UVO setting.
> |                 |          |         |         |       |       |       |
> |:----------------|:---------|:--------|:--------|:------|:------|:------|
> | Method          | Backbone | AR\_100 | AP\_100 | AP\_S | AP\_M | AP\_L |
> | BCnet           | R-50     | 36.28   | 20.87   | 6.42  | 13.92 | 31.27 |
> | Mask\_transfine | R-50     | 36.41   | 21.51   | 6.71  | 14.20 | 32.12 |
> | PointRend       | R-50     | 37.02   | 20.40   | 8.99  | 15.81 | 34.57 |
> | B-MaskRCNN      | R-50     | 36.21   | 20.40   | 5.98  | 13.61 | 30.69 |
>
> [a] and [b] are both excellent classical methods on weakly supervised segmentation. But their experimental settings  are not quite the same as ours, the comparison maybe unfair. Specifically, the semi-supervised setting in [a] and [b] assumes that the training set C = $A\cup B$, where examples from the categories in $A$ have masks, while those in $B$ have only bounding boxes.
>
> However, our semi-supervised setting is indeed as describe in section 3.5: Given a labeled set $D_l$ with both mask and box annotations, and an unlabeled set $D_u$ without any annotation, our goal is to train an OWIS model by leveraging both a large amount of unlabeled data and a smaller set of labeled data. The unlabeled data $D_u$ has neither mask nor bounding box annotation.
>
> Therefore we have made discussions of the difference between the settings of [a][b] and our methods, adding this part in Section 4.5 to the revised paper.
>
> *Reference:*
>
> *[a] Learning to segment every thing. CVPR, 2018.*
>
> *[b] Shapemask: Learning to segment novel objects by refining shape priors. ICCV, 2019.*
>
> *[c] PointRend: Image Segmentation as Rendering. CVPR, 2019.*
>
> *[d] Deep occlusion-aware instance segmentation with overlapping bilayers. CVPR, 2021.*
>
> *[e] Boundary-preserving Mask R-CNN. ECCV, 2020.*
>
> **Q7: Figure 5 is blurry.**
>
> **A7:** Thanks a lot. We are  sorry for the blurry figure and have replaced it according to the Reviewer’s comments.

---

> > ### Comment · Reviewer_THoq · 2022-11-22
> > **Incorrect Reference Format and Partial Concern Addressed**
> >
> > After reading the rebuttal and additional experiment results, part of my concerns has been addressed. The cross-task consistency loss did improve the open-world instance segmentation performance. Thus, I increased the rating. However, the paper still contains obvious reference format mistakes, many papers such as Mask2Former and "Learning to segment every thing" papers etc. has no any reference venues/sources.

---

> > > ### Author Response · Authors · 2022-11-25
> > > **Response to Reviewer THoq**
> > >
> > > Thank you again for your positive and constructive comments and suggestions on our manuscript. We are very sorry for our negligence of  reference format mistakes.  We have revised latex source code, correcting the irregular references of Mask2Former，MaskX-RCNN,  BCnet, Shapemask, Mask-Transfine and PointRend. We will update it in the final version.

---

> ### Author Response · Authors · 2022-11-19
> **Response to Reviewer THoq (Q1-Q4)**
>
> **We appreciate for Reviewers’ warm work earnestly, and hope that the correction will meet with approval.**
>
> **Q1: It lacks of sufficient validation by either theoretical interpretation or experimental evidence how the cross-task consistency loss works for providing an error-correcting mechanism for handling noisy annotations.**
>
> **A1:**  In our ICLR2023 manuscript submitted, we have not use words like “error-correcting mechanism” and "noisy annotations". We claim that " We propose a novel cross-task consistency loss that mitigates the issue of incomplete mask annotations." Some evidence and theoretical analysis of our method are summarized as follows：
>
> *Experimental evidences:*
>
> (1). In the COCO(VOC)$\xrightarrow[]{}$COCO(noneVOC) setting, the categories in training and test set have no overlap. Our proposed method improves the Mask2Former for about 19.87\% in $AR_{100}\%$ performace, which proves the proposed method has advantages on  discovering novel object.
>
> (2). Figure 1(g) shows the $AR_{100}\%$ of our method vs. baseline Mask2Former on COCO with partial annotation. From right to left, with the total number of classes decreases (i.e. more classes of instance annotations missed), the gain of our SOIS over baseline becomes larger. This can also severs as an evidence that our model can alleviate the incomplete annotations.
>
> (3). Visual results in Figure 8 in supplementary prove that the proposed method can discover many novel objects that never been annotated on the training data COCO2017-train.
>
> *Theoretical interpretation* is given in the Section3.3. Consistency loss enjoys the following appealing properties. It is self-calibrated and independent with the incompleteness level of labels. As shown in Figure 3, for a instance mistakenly annotated as background, but the foreground prediction branch and mask prediction branch both correctly find it, the model would be punished through mask loss and foreground loss. However, the consistency loss think this prediction is correct. In this way, consistency loss down-weights the adverse effects caused by other unreliable segmentation loss. The mitigation and the compensation factor synergize to relieve the overwhelming punishments on unlabeled instances.
>
> **Q2:How does the “cross-task consistency loss” help to handle and separate neighboring occlusion objects**
>
> **A2:**	Note that our paper does not focus on the problem of occlusion, where only has partial mask of an instance. In contrast, The "incomplete" in our paper means the whole instance mask is missed. In the future, it would be interesting to look into this new problem of occlusion.
>
> **Q3: The tech novelty of the paper is weak and incremental. Being first to adapt Mask2Former to OWIS is an incremental idea.**
>
> **A3:** Thanks for you concern. Towards the incomplete annotation issue and pursing more efficient network architecture, we propose our novel framework. The novelty of our proposed method is as follows:
>
> First, our proposed cross-task consistency loss is simple but efficient. It is designed to handle incomplete annotation problem. For an instance mistakenly annotated as background, but the foreground prediction branch and mask prediction branch both correctly find it, the model would be punished through mask loss and foreground loss. However, the consistency loss thinks this prediction is correct. In this way, consistency loss down-weights the adverse effects caused by other unreliable segmentation loss. The mitigation and the compensation factor synergize to relieve the overwhelming punishments on unlabeled instances.
>
> In this way, it easily helps extend CWIS methods to gain better performance on OWIS task. Experimental results show the advantages of our method. It improves the baseline Mask2Former for 14.27\% and 14.38\% on AP100 in UVO−→UVO setting and COCO−→UVO setting, respectively.
>
> Second, Mask2Former is chosen as baseline to overcome the incomplete annotation after in-depth analyse, but not an incremental work. The reasons are as follows:
>
> Mask2Former generates $N$ binary masks corresponding to $N$ queries and then matches these masks against the ground truth via the Hungarian matching method. During this process, only masks matched to the ground-truth objects are supervised. The unmatched masks will not incur any losses. Also, the background region in the generated mask is only “background for a particular instance”. Object instances could still be generated from those background pixels from other masks, including those masks that are not matched to the ground-truth object. Thus, the unannotated object could emerge from the masks that are not matched to the existing annotations without being penalized.
>
>
> **Q4: It should be called query-based or transformer-based, instead of one-stage.**
>
> **A4:** Thanks for the kind suggestion, we agree with the comment and rename the method in the revised manuscript.

---

### Official Review · Reviewer_2Nw5 · 2022-10-25

**Confidence:** 3
**Clarity, Quality, Novelty And Reproducibility:** 1. The paper is well written, and alo…
**Correctness:** 3
**Technical Novelty And Significance:** 2
**Empirical Novelty And Significance:** 3
**Recommendation:** 5

**Strength And Weaknesses:**

Strength:
1. This paper is easy to follow since the core idea is presented clearly.
2. The method can help address the missing annotation problems in the public benchmark datasets and further boost the accuracy of SOTA.
3. The method can enhance the model performance without introducing further model parameters.
4. The method is compatible with utilizing unlabeled images for semi-supervised learning.

Weakness:
1. Since the single-stage instance segmentation is from the Mask2Former, the core contribution seems to become cross-task consistency regularization loss term only.
2. There are a few typos, replicated citations, and redundant spacing in the context.


**Summary Of The Paper:**

This work leverage the Mask2Former for single-stage instance segmentation plus proposes cross-task consistency regularization to achieve the class-agnostic segmentation of all objects in an image. Specifically, the proposed method uses a foreground prediction branch, similar to the idea of saliency prediction, to help the model learn the objects in an image, then encourage the sum of the predicted objects' map to resemble the foreground map. The proposed approach effectively alleviates the incomplete annotation problem in common datasets, furthermore, it is proven to outperform SOTA models.

**Summary Of The Review:**

The idea of encouraging the consistency between foreground prediction and instance mask prediction is good, it helps address the long-existing problems in the benchmark datasets and even helps alleviate the annotation burden in the future. Also, the experiments show the proposed method is effective and help boost the performance over the state-of-the-art. However, I am unsure if the consistency loss framework is significant enough to outstand peers.

---

> ### Author Response · Authors · 2022-11-18
> **Response to Reviewer 2Nw5**
>
> **Thank you very much for your careful review and constructive comments with regard to our manuscript.**
>
> **Q1: I am unsure if the consistency loss framework is significant enough to outstand peers.**
>
> **A1:** Here, we re-emphasise our significance:
>
> *Problem importance:* OWIS setting is more in line with practical industrial application scenarios, such as autonomous driving. And it is also more challenging compared to the traditional CWIS task.
>
> *Superior performance:* Experimental results show the advantages of our method. It improves the baseline Mask2Former for 14.27\% and 14.38\% on $AP_{100}$ in UVO$\xrightarrow{}$UVO setting and COCO$\xrightarrow{}$UVO setting, respectively.
>
> *Simple yet effective design:* We proposed a practical method. Because it is simple yet effective, likely many people will follow and adopt it. Maybe it can become a standard approach, also generalize to related research field, like open world semantic segmentation.
>
> **Q2: There are a few typos, replicated citations, and redundant spacing in the context.**
>
> **A2:** We double checked the paper, and corrected the typos, replicated citations, and redundant spacing.

---

> > ### Comment · Reviewer_2Nw5 · 2022-11-23
> > **Response to author feedback**
> >
> > Thanks for providing the feedback. I understand the proposed consistency regularization can help deal with missing annotation and help boost the generalization performance to a much larger extent; however, I wonder how much the foreground prediction's performance will affect the performance of instance segmentation. In other words, how does this impact consistency regularization if the foreground prediction is not accurate or noisy? Thanks.

---

> > > ### Author Response · Authors · 2022-11-25
> > > **Response to Reviewer 2Nw5**
> > >
> > > **A:** Thank you for raising this insight question. Following the reviewer's comments, we added some new experiments and provided some in-depth analyses on existing experiment results.
> > >
> > > **1.** First, we evaluated the accuracy of the foreground prediction in our model. Specifically, in the COCO-->UVO setting, we first generated the foreground prediction groundtruth by computing the union of mask groudtruth. **Then we tested the accuracy of the foreground predictions on images from the UVO-val dataset. Its AP(Average Precision) and AR (Average Recall) are 68.6% and 74.5%, respectively.** In other words, foreground prediction results are not completely accurate and do contain noises. Under this circumstance, our proposed method still achieves the AP100 and AR100 of 32.21% and 54.86%（The AP100 and AR100 of baseline Mask2Former with no forground and consistancy loss are 28.16% and 51.38%, respectively), indicating that the **proposed method is robust to the noise in the foreground predictions.**
> > >
> > > **2.**  From another perspective, some experimental results in our revised paper can further reflect **how foreground predictions with different noises affect the OWIS results**. Here, we describe the details of this experiment and provide some analysis.
> > >
> > > **(1)**  We first **generate two foreground predictions with different levels of noise**. Specifically, we train our proposed method with 30% and 50% of labeled data from the UVO dataset, denoting the models as  $LO_{30}-TOIS$ and $LO_{50}-TOIS$, respectively. Obviously, the foreground predictions from $LO_{50}-TOIS$ should be more accurate and less noisy than those from $LO_{30}-TOIS$.
> > >
> > > **(2)**  We further **train $LO_{30}-TOIS$ and $LO_{50}-TOIS$ model on the unlabeled data.** During this process, foreground predictions serve as the only source of constraints. Specifically , $LO_{30}-TOIS$ is continued to be trained on the remaining 70% of unlabeled data in UVO to obtain model $Semi-TOIS_{30}$. $LO_{50}-TOIS$ is continued to be trained on the remaining 50% of unlabeled data in UVO to obtain model $Semi-TOIS_{50}$. (For fairness, the total amount of unlabeled and labeled data should be the same in both sets of experiments.) During this training process, there is no mask groundtruth, and the **mask predictions are constrained only by the foreground prediction through the consistency loss**. By comparing the performance improvement of $LO_{30}-TOIS$--> $Semi-TOIS_{30}$ with that of $LO_{50}-TOIS$--> $Semi-TOIS_{50}$, we can evaluate how much these two foreground predictions (with different accuracy and levels of noise) affects the final OWIS prediction results.
> > >
> > > **(3)**  As shown in Figure 5 in the main paper, compared to those of $LO_{30}-TOIS$, the AP100 and AR100 of $Semi-TOIS_{30}$ are improved by 3.36% and 5.33%, respectively, Compared to those of $LO_{50}-TOIS$, the AP100 and AR100 of $Semi-TOIS_{50}$ are improved by 2.36 % and 6.12%, respectively. The performance improvements brought by applying foreground predictions with different accuracy are comparable.
> > >
> > > Through the above experiments, we can conclude that: **(1）The proposed model can achieve significant OWIS results in the presence of certain foreground prediction noises. **(2)** The improvements on OWIS performace show certain robustness for the changes on foreground prediction accuracy. This also indicates that "noise" and "not accuracy" of foreground prediction (within a certain range of non-extreme) can be tolerated in the proposed method.**
> > >
> > > We greatly appreciate the efficient and professional comments. And we hope the response will address your concerns. Very pleased to hear from you in the comments and suggestions.

---

> > > ### Author Response · Authors · 2022-12-11
> > > **Response to Reviewer 2Nw5**
> > >
> > > We deeply appreciate your time and consideration of our paper. We have provided some experimental results and statistical analysis to answer this question in the response. We are highly wondering whether this response has addressed your concerns. And If you have additional questions, we would be very happy to answer them before tomorrow's deadline.

---

### Official Review · Reviewer_x3r9 · 2022-11-03

**Confidence:** 3
**Correctness:** 3
**Technical Novelty And Significance:** 2
**Empirical Novelty And Significance:** Not applicable
**Recommendation:** 6

**Clarity, Quality, Novelty And Reproducibility:**

Clarity - Could be improved. The figure and caption do not complement each other. Also, there are notations in the paper that are not clearly defined (Section 3.2) What is Ki? The number of annotated object instances in a given image i? Should be explicitly specified.

Quality - The paper could be improved in terms of language and structure - there are also a number of typos and missing references that should be addressed. Most parts of Figure 1 are not referenced in the paper.

Novelty - I do not consider the contribution novel, but an extension of an existing state-of-the-art that was applied in a different learning setup that deals with missing annotations - not the fully-supervised case. For this purpose alone the authors apply a consistency loss computed as a segmentation loss between the instance segmentation branch and a foreground prediction branch and use this as a constraint in learning when labels are missing.

Reproducibility - The authors have released the code in their supplementary material - the experiments should be easy to reproduce.

**Strength And Weaknesses:**

Strengths:
* The tackled problem is challenging and of interest to the research community - open-world instance segmentation is far from being solved - since it implies the discovery of novel object instances that were not seen during training

Weaknesses:
* Problem statement description could be improved. The introductory section does not properly address the tackled task or the way in which this work contributes to current issues.
* Hard to grasp from the paper what are the authors' contributions - I did not find the paper an easy read.
* The authors rename the Mask2former architecture as SOIS and wrongfully consider it their contribution since the architecture itself did not change as the authors stated themselves but rather the scenario in which it was used (extension to a semi-supervised scenario by employing a proxy constraint, in the form of a consistency loss between predicted masks of two existing branches in the architecture)

Other comments:
* Benchmarks are not referenced when first mentioned - same goes for COCO and UVO in the abstract - at least mention that these are datasets.
* Figure 1 is crowded and hard to follow all of its parts. It would make more sense to split it into two parts - a, b, c - Part 1 and d, e, f, g - Part 2. In (b) consistency is misspelled and Cityscapes is misspelled in the figure caption.
* Figure 1c, 1d, 1e and 1f are not referenced in the paper.
* Figure 2 is not informative at all - What is the purpose of that backbone? - What is the format of the output? What is the input of the FCN in the foreground prediction branch and why not use the RGB image? The image caption does not complement the figure - most of the notations are not explained and the flow of the different branches is very hard to follow. Also, sentences such as "the foreground prediction branch segments a foreground region" can be derived from the naming of these components. Authors should highlight only the information that cannot be inferred solely from the naming.

**Summary Of The Paper:**

The paper tackles the task of open-world instance segmentation in images (no predefined set of classes). The proposed method extends on an existing state-of-the-art single-stage (without employing a dedicated proposals procedure) approach, such as Mask2former, through the addition of a foreground segmentation branch. The foreground prediction map is used as an auxiliary supervisory signal for the instance prediction branch to increase the consistency among these maps. The consistency loss applied on top of these predictions also alleviates the problem of incomplete instance annotation which could be considered a core contribution of this work (working with labeled and unlabeled data simultaneously - semi-supervised scenario).

**Summary Of The Review:**

Besides the aforementioned issues with the current submission, I do not find it fit for publication not just for the structure or lack of clarity, but mostly due to its incremental contribution to existing works.

---

> ### Author Response · Authors · 2022-11-18
> **Response to Reviewer x3r9 (Q6-Q10)**
>
> **Q6: Figure 2 is not informative at all - What is the purpose of that backbone? - What is the format of the output?**
>
> **A6:** Backbone is utilized to extract features from input images. Here we use pre-trained Resnet-50 and Swin-base model. This choice follows the mainstream approaches in instance segmentation. Using the same backbone with compared methods makes it fair to do the evaluation. Backbone generates an image feature map $F\in C\times \frac{H}{S} \times \frac{W}{S}$ , where $C$ is the number of channels and $S$ is the stride of the feature map ($C$ depends on the specific backbone and we use $S = 32$ in this work). $H$ and $W$ represent the height and weight of original image, respectively.
>
> **Q7:What is the input of the FCN in the foreground prediction branch and why not use the RGB image? The image caption does not complement the figure - most of the notations are not explained and the flow of the different branches is very hard to follow.**
>
> **A7:** As shown in Figure 2, the inputs of the foreground prediction branch is the feature generated from the backbone. The input image is indeed an RGB image from classic instance segmentation dataset: **Cityscapes dataset**.
>
> **Q8: Sentences such as "the foreground prediction branch segments a foreground region" can be derived from the naming of these components. Authors should highlight only the information that cannot be inferred solely from the naming.**
>
> **A8:** Thank you for the suggestion. We agree that some sentences, including the one you point out, are redundant. However, different people have different opinions on the same writing style (which is actually what happened in the reviews of this paper). We believe that explaining the same thing in different phrases, while perhaps redundant, makes it more readable to a wider audience.  This detailed description allows every reader, with or without a professional background in related research field, to better understand the aim of each branch.
>
> Other information, like the entire workflow of the framework is presented in the main text. Due to the limited space, we have not repeated it in the caption.
>
> **Q9: There are notations in the paper that are not clearly defined (Section 3.2) What is Ki? The number of annotated object instances in a given image i? Should be explicitly specified.**
>
> **A9:** Thanks for pointing this out, we have added the corresponding explanation. $K_i$ means the number of annotated object instances in a given image $i$.
>
> **Q10: This work is to deals with missing annotations - not the fully-supervised case. For this purpose alone the authors apply a consistency loss computed as a segmentation loss between the instance segmentation branch and a foreground prediction branch and use this as a constraint in learning when labels are missing.**
>
> **A10:** We would like to make a clarification here:
>
> Incomplete annotation or missing annotation is an **inherent problem** in the OWIS task. Because it is impossible to label all the instances in an image completely in the open world setting. **Although in the Fully-supervised setting, this problem also exists.** Specifically, Fully-supervised setting is to use all annotated images and the corresponding annotations in the training dataset. Note that these annotations may not cover all objects in the image, as shown in Figure 1 and in Figure 8 ( in supplymentary ). We never manually remove any annotations in this setting.

---

> ### Author Response · Authors · 2022-11-19
> **Response to Reviewer x3r9 (Q1-Q5)**
>
> **Thanks to the reviewer for helpful comments.**
>
> **Q1: Problem statement description could be improved. The introductory section does not properly address the tackled task or the way in which this work contributes to current issues.**
>
> **A1:** Thank you for the suggestion. We have updated the Introduction as summarized below. We are happy to update more if you have more specific suggestions.
>
> Summary of our introduction：
>
> Paragraph 1: Traditional CWIS method can only segment a finite set of predefined classes, which can not satisfy many real-world applications. This gives rise to OWIS task that aims to segment all instances.
>
> Paragraph 2: SOTA methods rely on open-world detection boxes and we choose one-stage Mask2Former as baseline.
>
> Paragraph 3: Challenges in OWIS task: Incomplete annotations.
>
> Paragraph 4: Shortcoming of SOTA.
>
> Paragraph 5: Introduce our work TOIS.
>
> Paragraph 6: Our method can be extend to semi-supervised learning.
>
> Paragraph 7: Our contributions: new OWIS framework, novel consistency loss and superior performance.
>
> **Q2:Hard to grasp from the paper what are the authors' contributions - I did not find the paper an easy read.**
>
> **A2:** Thank you for the comments. Although other reviewers have different opinions (R2:"The paper is well written, and along with the figures, it is easy to follow overall." R3: "Good paper structure." R4:"The method is well described."), we value your opinion and would like to make sure that our writing is clear to everyone. The specific part mentioned is "Hard to grasp from the paper what are the authors' contributions" so we tried to improved the way we introduce the contributions in the Introduction. Here we claim our contributions. If you have more specific comments, we will update more.
>
> Contributions:
>
> (1) Based on the Mask2Former, we design the first single-stage transformer-based open-world instance segmentation framework.
>
> (2) Incomplete annotation issue is the real-world challenge in OWIS task. Current fully-supervised models witness performance drop due to such issue. Towards this end, we propose the novel cross-task consistency loss in our framework.
>
> (4) Extensive experiments demonstrate our model achieves leading performance under the fully-supervised setting .
>
> (3) We further extend the proposed cross-task consistency loss into semi-supervise setting to explore the benefit of unlabeled data ( images with no annotation).  Experiments have demonstrated the effictiveness of our method.
>
>
>
>
> **Q3: The authors rename the Mask2former architecture as SOIS and wrongfully consider it their contribution since the architecture itself did not change as the authors stated themselves but rather the scenario in which it was used (extension to a semi-supervised scenario by employing a proxy constraint, in the form of a consistency loss between predicted masks of two existing branches in the architecture).**
>
> **A3:** Sorry, it is not our intention to claim that the Mask2former architecture is our contribution. Our contribution is to develop ways to apply Mask2Former for open-world instance segmentation. We have rewrited the introduction to make this point clearer, as summarized in **A1**. We do use Mask2Former as baseline, but our method provides additional value and outperforms the straightforward adaptation of Mask2Former into open-world instance segmentation.
>
> As shown in  Table 1 and Table 3, our proposed consistency loss is clearly efficient for both fully-supervised and semi-supervised OWIS. In fully-supervised setting, our method improves the baseline Mask2Former for 14.27\% and 14.38\% on $AP_{100}$ in UVO$\xrightarrow{}$UVO setting and COCO$\xrightarrow{}$UVO setting, respectively. In semi-supervised setting, our method improves the $AP_{100}$ of Mask2Former by 29.40\%  when both methods are trained with 50\% labeled training data, as shown in Table 8.
>
>
> **Q4: Benchmarks are not referenced when first mentioned - same goes for COCO and UVO in the abstract - at least mention that these are datasets.**
>
> **A4:** We fixed them. Thank you for the suggestion.
>
> **Q5: Figure 1 is crowded and hard to follow all of its parts. It would make more sense to split it into two parts - a, b, c - Part 1 and d, e, f, g - Part 2. In (b) consistency is misspelled and Cityscapes is misspelled in the figure caption. Figure 1c, 1d, 1e and 1f are not referenced in the paper.**
>
> **A5:** We agree that the presentation of Figure 1 could be improved. We have added bounding boxes to divide the Figure 1 into two parts to make it clearer. In fact, the Figure 1(d-f) have been cited in the last paragraph of Introduction. In the revised paper, Figure 1(c) has been cited and typeos have been corrected. If you have more suggestions, we are happy to apply them.

---

### Decision · Program_Chairs · 2023-01-20

**Decision:**

Reject

**Justification For Why Not Higher Score:**

Given the limited contribution of the article, I recommend rejecting it.

**Justification For Why Not Lower Score:**

N/A

**Metareview: Summary, Strengths And Weaknesses:**

This article addresses the problem of open-world instance segmentation with no predefined classes. The problem is interesting, challenging, and important. The reviewers agree that the article is straightforward, easy to follow, boosts state of the art, does not introduce new parameters, and is compatible with semi-supervised learning.

However, they also agree that the novel contribution is very limiting since it is a straightforward extension of Mask2former where a foreground segmentation branch is added and a cross-task consistency loss. They also highlight that the problem statement and the intro can be improved. They didn't clarify what part of the contribution is from them and what comes from Mask2former. The article overstates the contributions of the method. It also misses comparison with other state-of-the-art articles and other minor comments.

The authors addressed some of the comments, and the paper improved. However, given the limited contribution of the article, I recommend rejecting it.